# *VvERF111* Regulates Chlorophyll Degradation by Activating Expression of *VvCLH1*, Leading to Rachis Browning in Grape

**Dongfang Zou** [†], **Jingwen Li** [†], **Xia Ye, Xianbo Zheng, Bin Tan, Jun Cheng, Wei Wang, Zhiqian Li** *
**and Jiancan Feng** *

College of Horticulture, Henan Agricultural University, 95 Wenhua Road, Zhengzhou 450002, China
* Correspondence: zhiqianli@henau.edu.cn (Z.L.); jcfeng@henau.edu.cn (J.F.)
† Those authors contributed equally to this work.

**Abstract:** The plant growth regulator ethylene influences rachis browning in grape (*Vitis vinifera* L.). Although the ethylene signaling pathway is well defined, there is limited knowledge on its mode of action during rachis browning. Here, we show that an ethylene response factor (*VvERF111*) positively regulates chlorophyll degradation in rachis by binding to a DRE motif in the promoter of *VvCLH1*. The expression of *VvERF111* and *VvCLH1* in rachis was induced by ethylene and inhibited by 1-methylcyclopropene (1-MCP). VvERF111 belongs to the ERF IX subfamily of the APETALA2/ethylene responsive factor (AP2/ERF) superfamily, shows transcriptional activity in yeast, and is localized in the nucleus and membrane. The transient overexpression of *VvERF111* or chlorophyllase (*VvCLH1*) in grape leaves accelerated chlorophyll degradation. In *VvERF111*-overexpressing leaves, transcript levels of *VvCLH1* were also increased. Our findings offer a deeper understanding of the transcriptional regulation of chlorophyll degradation during the rachis browning of grape.

**Keywords:** chlorophyll breakdown; ethylene; *VvERF111*; transcriptional regulation

## 1. Introduction

Fruit and stem color are highly visible indicators of fruit quality to consumers. Grape is cultivated, shipped, and consumed worldwide. After harvest, the storage and shipping of table grape must be carried out in a way that maintains the overall quality of the bunches. The priority is the prevention of berry decay, but rachis browning must also be considered [1]. To the consumer, green main stems and individual rachis to each berry are indications of freshness, while brown rachis can be a major cause of consumer rejection and, eventually, fruit waste. Rachis browning is also a big problem during post-harvest storage. Water loss is one of the key factors in rachis browning [2]; however, rachis quality also varies during prolonged storage under high relative humidity [2]. This indicates that there are additional factors participating in rachis browning.

Chlorophyll is the key pigment for photosynthesis, which most living plants depend on. Chlorophyll is subject to both catabolic and anabolic metabolism. The color change of leaves in autumn and of fruit during ripening are conspicuous examples of the degradation of chlorophyll [3–5]. Chlorophyll degradation-related genes and enzymes have been studied in a range of plant species and include chlorophyllase (CLH), oxygenase (PAO), pheophytin pheophorbide hydrolase (PPH), RCC reductase (RCCR), and yellowing1/stay-green1 (NYE1/SGR1) [6–8]. CLH is a rate-limiting enzyme during chlorophyll breakdown [9]. Transgenic broccoli (*Brassica oleracea* var. *italica*) expressing an anti-sense chlorophyllase (*BoCLH1*) shows delayed post-harvest yellowing [10]. Chlorophyll catabolism has not been well-studied in grape rachis, but it is known that chlorophyll content is reduced and the expression of chlorophyll degradation genes is increased during rachis browning [11,12].

Ethylene is one of the causes of rachis browning. Although grape is considered a non-climatic fruit that does not exhibit a large rise in ethylene production or respiration rate,

a study showed that grape rachis can produce more ethylene than grape berries during fruit ripening [13], which leads to rachis browning. Furthermore, ethylene treatment enhances rachis browning, while treatment with the ethylene action inhibitor 1-MCP inhibits it [14,15]. 1-MCP treatment also suppresses the expression of chlorophyll degradation genes in the skin of pears [16]. Chlorophyll degradation can be enhanced by ethylene treatment [17].

The ethylene response factor (ERF) is transcription factors within the ethylene signaling pathway that is associated with chlorophyll levels, either by increasing its degradation or by suppressing its synthesis [18]. The ERF members recognize GCC-box *cis*-acting elements (AGCCGCC) and DRE motifs in the promoters of ethylene-responsive genes to regulate their expression [4,5]. For instance, *AtERF4* binds directly to the *AtCLH1* promoter and increases its expression [19]. *AtERF72* directly binds to the *AtCLH1* promoter under iron deficiency [11]. *CitERF6* and *CitERF13* in citrus and *MdERF17* in apple regulate chlorophyll degradation during the degreening of the fruit peel [4,5,20,21]. Last year, we reported that *VvERF95* participates in grape rachis browning by regulating the expression of *VvPAO1* [12], while another group showed that *VvERF17* regulates chlorophyll degradation in grape berry skin [17]. In tomato, *SlERF.J2* regulates expression of a chlorophyll synthesis gene [18]. Together, these reports suggest that ERF transcription factors have conserved functions in chlorophyll catabolism. In our previous study, *VvERF111* was differentially expressed in the grape cluster, showing higher expression during rachis browning [22] and a negative correlation with chlorophyll content. These preliminary results indicated that *VvERF111* has a potential role in chlorophyll degradation during rachis browning.

In order to understand the role of *VvERF111* during rachis browning, the transcript levels, protein localization, and functional activity of *VvERF111* were characterized. We demonstrated that the ERF IX subfamily gene *VvERF111* contained a conserved AP2/ERF domain, showed transactivation activity in yeast cells, and localized to the nucleus and membrane. VvERF111 could bind to the DRE motif in the promoter of the chlorophyll degradation gene *VvCLH1*, to up-regulate its expression, which resulted in accelerated degradation of chlorophyll. The data presented here furthers our deciphering of the mechanisms of rachis browning in grape.

## 2. Materials and Methods

### 2.1. Plant Materials

Five-year-old vines of the grape cultivar 'Shine-Muscat' (*Vitis labruscana* Baily × *Vitis. vinifera* L.) were maintained in a commercial vineyard (Zhengda Vineyard, Zhengzhou City, Henan, China) under normal growth conditions. Clusters of similar size and maturity were selected and used for 1-MCP treatment, as we previously reported [12]. Three biological replications were carried out for the whole experiment.

### 2.2. Gene Isolation and Sequence Analysis of VvERF111

The full-length coding sequence (CDS) of *VvERF111* was cloned from Shine Muscat using primers listed in Table S1. Sequences of ERF genes in *Arabidopsis* and homologous genes were downloaded from the TAIR and NCBI databases. Sequence alignments and phylogenic trees were generated using MEGA7 according to a previous report [23]. Promoter sequences (1500 bp) upstream of the ATG start codon of chlorophyll degradation genes were download from the grape genome (12 X). The DRE motif (A/TCCGAC) was manually queried in the promoter of *VvCLH1*.

### 2.3. Transcriptional Activation Activity

The CDS of *VvERF111* was inserted into the vector pGBKT7 (BD) using the NdeI and PstI restriction sites, to yield *pGBKT7-VvERF111*. The empty *pGBKT7* vector was used as a control. *pGBKT7* and *pGBKT7-VvERF111* were transformed into yeast strain Y2HGold and then grown on SDO (Single Dropout, SD-Trp) for three days. Colonies were spot plated onto SDO, SDO/A, and SDO/A/X to observe yeast growth. Primers used in this study are listed in Table S1.

### 2.4. Subcellular Localization Analysis

The CDS of *VvERF111* (without the termination codon) was introduced into the *pSAK277-GFP* expression vector digested with *Eco*RI under the control of the CaMV 35S promoter. The constructs *35S-VvERF111-GFP* and *pSAK277-GFP* were introduced into *A. tumefaciens* GV3101, which was then transformed into *Nicotiana benthamiana* leaves. After incubation for 48 h, the fluorescence was detected by laser confocal microscopy.

### 2.5. Yeast One-Hybrid Analysis

The CDS of *VvERF111* was cloned into the pB42AD vector in restriction enzyme sites EcoRI/XholI, and promoter sequences of the chlorophyll degradation genes *VvSGR, VvSGRL, VvCLH1, VvPAO1, VvRCCR,* and *VvNOL* were inserted into the pLaczi vector ligated into the Placzi vector in restriction enzyme sites KpnI/SalI. The *VvERF111* and promoter sequence constructs were co-transformed into yeast EGY48 [5,24]. The transformants were cultured on SD/-Trp/-Ura and on SD-Gal/Raf-X-gal solid media, and observed after 2 to 3 days. To further identify which *cis*-element in the promoter of *VvCLH1* can be bound by VvERF111, the promoter sequence of *VvCLH1* was divided into three segments. Each segment was tested for activation by VvERF111, and the segment that could interact with VvERF111 was further divided into six segments and tested; the mutation of DRE mitif was conducted in segment to verify if it was the key cis-element bound by VvERF111. All primers were constructed as shown in Table S1.

### 2.6. Dual Luciferase Assay

The *VvCLH1* promoter region was inserted into the pGreenII0800-LUC vector digested with *KPN*I and *NCO*I to yield the reporter construct [25], and the CDS of *VvERF111* was introduced into pSAK277 to yield the effector construct, with pGreenII0800-LUC and GFP used as negative controls. The constructs carrying *VvERF111* and the promoter of *VvCLH1* were injected into *N. benthamiana* leaves [26]. LUC and REN luciferase activities were determined with a dual luciferase reporter kit (Dual-Luciferase Reporter Assay System, Promega, Madison, WI, USA). Each reaction was performed in six replicates.

### 2.7. Transient Overexpression of VvERF111 in Grape

*35S-GFP*, *35-VvERF111-GFP*, and *35S-VvCLH1-GFP* were individually transformed into *A. tumefaciens* GV3101. Fully expanded third leaves (4 weeks old) were removed from in-vitro-grown grape plants and transiently transformed with vacuum infiltration as described [22,27]. The expression of *VvCLH1* and *VvERF111* in transformed leaves was determined by qRT-PCR. Phenotypes were observed after 3 days, and the chlorophyll contents were quantified [12] in *VvERF111*- and *VvCLH1*-overexpressing leaves, with leaves transformed with *35S-GFP* used as a control. This experiment was biologically replicated 3 times, and 20 leaves were selected for each experiment.

### 2.8. RNA Extraction and Gene Expression Analysis

Total RNA from leaves and rachis were extracted using the RNA kit (Quick RNA Isolation). First strand cDNA was synthesized with a reverse transcription kit (RT Master Mix for qPCR II). Real-time quantitative polymerase chain reaction (RT-qPCR) was performed using SYBR Green and an Applied Biosystems 7500 FAST RT-PCR system. The grape GAPDH (CB973647) gene was used as an internal control. The experiment was conducted in three biological replicates. The $2^{-\Delta\Delta CT}$ method [28] was used to calculate the relative gene expression levels. Details on the gene-specific primers that were used are shown in Table S1. ANOVA was used to identify statistically significant differences among genes ($p < 0.05$).

## 3. Results

### 3.1. VvERF111 Belongs to the ERF IX Subfamily and Has a Conserved AP2/ERF Domain

ERF transcription factors have pivotal roles during chlorophyll degradation. We previously found that *VvERF111* transcript levels increased in browning rachis during storage. Treatment with the ethylene action inhibitor 1-MCP inhibits the expression of *VvERF111* [22] and increases the chlorophyll content in rachis treatment [12], which indicates that *VvERF111* may play a potential role during rachis browning of grape. The ERF transcription factor family is classified into 12 subfamilies [29]. *Arabidopsis* genes representing each subfamily were downloaded from the TAIR database, and the protein sequences were used to construct a phylogenetic tree. VvERF111 belonged to the ERF IX subfamily and was most closely related to AtERF100/AtERF1 and AtERF101/AtERF2 (Figure 1A). VvERF111 was in the same subfamily as VvERF95, which we showed participates in the regulation of chlorophyll degradation [12]. Amino acid sequence analysis confirmed that *VvERF111* contained a conserved AP2/ERF domain (Figure 1B).

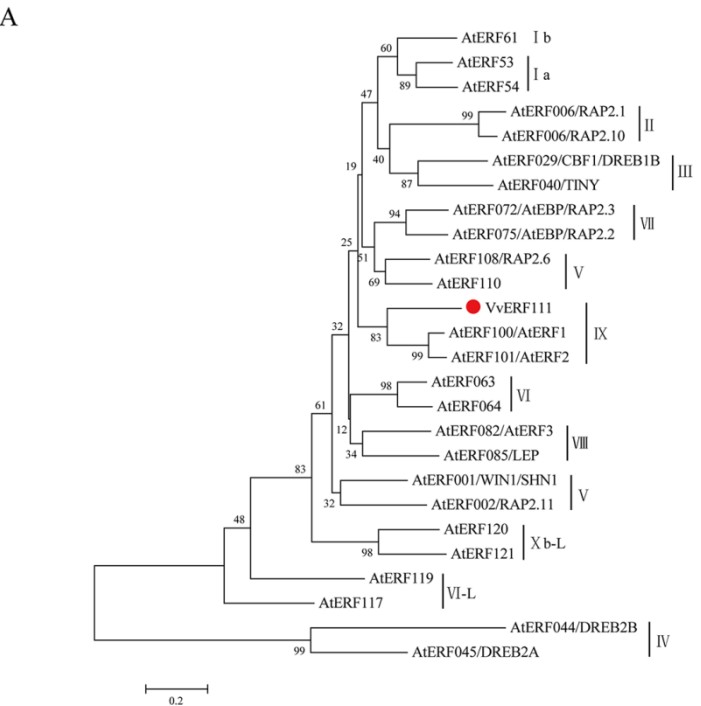

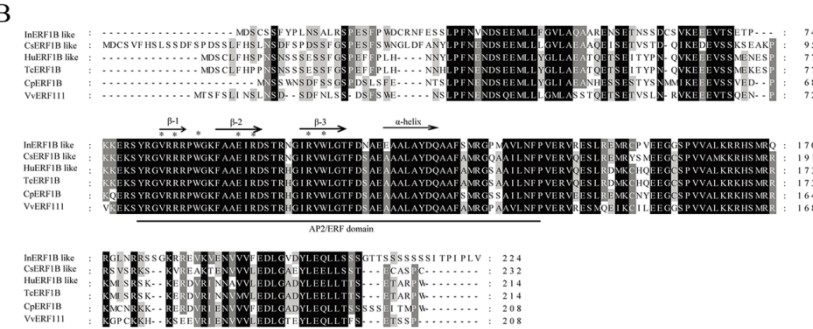

**Figure 1.** Sequence analysis of VvERF111. (**A**) Phylogenetic analysis of VvERF111 and the ERF proteins from *Arabidopsis thaliana* representing each of the 12 families. (**B**) Multiple amino acid sequence alignment of VvERF111 and homologous proteins from *Ipomoea nil* (XP_019181066), *Camellia sinensis* (XP_028082616), *Herrania umbratica* (XP_021290986), *Theobroma cacao* (EOY06019), and *Carica papaya* (XP_021899534). Black line underlined letters represent AP2/ERF domain. Predicted α-helix and β-sheet regions were denoted with black bar and arrows. Asterisks represent amino acid residues that directly make contact with DNA.

### 3.2. Expression of VvERF111 and VvCLH1 Induced by Ethylene

In Shine Muscat grape, the rachis gradually browns during post-harvest storage, and 1-MCP treatment can delay rachis browning, according to our previous reports [12,22]. The transcriptional response of *VvERF111* and *VvCLH1* in the rachis was examined following treatment with 1-MCP or ethylene at 0, 4, 12, 24, and 48 h. The transcript levels of *VvERF111* and *VvCLH1* increased after ethylene treatment and decreased after 1-MCP treatment (Figure 2A,B). This suggested that ethylene affected the expression of *VvERF111* and *VvCLH1*. The expression of *VvERF111* and *VvCLH1* were similar (Supplemental Table S2) [12]. The DRE motif, which binds ERF transcription factors, was found in the promoter of *VvCLH1*. This prompted the hypothesis that *VvERF111* regulates the expression of *VvCLH1*, thus participating in ethylene induced degradation of chlorophyll.

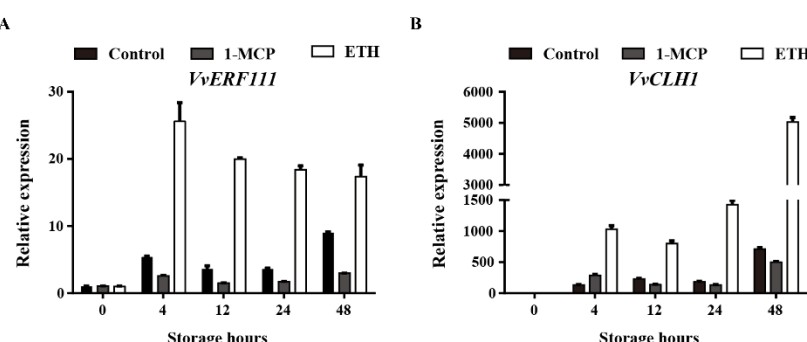

**Figure 2.** Expression of *VvERF111* (**A**) and *VvCLH1* (**B**) during rachis browning of grape and their responses to ethylene and 1-MCP treatments. Data represented mean ± SE from three independent biological replicates.

### 3.3. Analysis of VvERF111 Subcellular Localization and Transcriptional Activation Activity

To identify the molecular mechanisms of VvERF111-mediated Chl degradation, the ability of VvERF111 to serve as a transcription factor was assessed through determining its subcellular location and transcriptional activation properties. The CDS of *VvERF111* was introduced into the pGBKT7 vector. The pGBKT7- *VvERF111* construct was transformed into strain Y2HGold, with the empty pGBKT7 vector used as control. pGBKT7-*VvERF111* and pGBKT7 grew normally on SDO, but only pGBKT7-*VvERF111* could survive on SDO/A and turn blue on SDO/A/X (Figure 3A). This indicated that *VvERF111* could activate transcriptional in yeast.

A nuclear localization signal (NLS) signal is included in the VvERF111 sequence, indicating that VvERF111 maybe a nuclear protein. To confirm this speculation, the CDS of *VvERF111* was fused with GFP under the control of the CaMV35S promoter. The *35S-VvERF111-GFP* and *35S-GFP* constructs were transformed into *N. benthamiana* leaves. The green fluorescence of VvERF111-GFP was observed at both the cell membrane and in all nuclei stained with mCherry (Figure 3B, lower panels). The GFP only signal was detected in one nucleus and diffusely in the cell, including at the membrane (Figure 3B). The results suggested that VvERF111 was localized in both the nucleus and the membrane.

### 3.4. VvERF111 Activates Expression of the VvCLH1 Promoter

To investigate the ability of VvERF111 to bind to the promoter of *VvCLH1* and other chlorophyll degradation genes, the promoters of *VvSGR*, *VvSGRL*, *VvCLH1*, *VvPAO1*, *VvRCCR*, and *VvNOL* were fused into the pLaczi vector, and the CDS of *VvERF111* was fused into the pB42AD vector. The constructs were co-transformed in pairs into the yeast strain EGY48. The results indicated that *VvERF111* could bind the *VvCLH1* promoter (Figure 4A). Dual-luciferase reporter assays were carried out to test whether *VvERF111* was directly activating or suppressing the expression *VvCLH1*. The results showed that the LUC activity was significantly enhanced with co-tranformed of *VvERF111* and the *VvCLH1*

promoter compared to the control (Figure 4B), which indicated that *VvERF111* positively regulates the expression of *VvCLH1*.

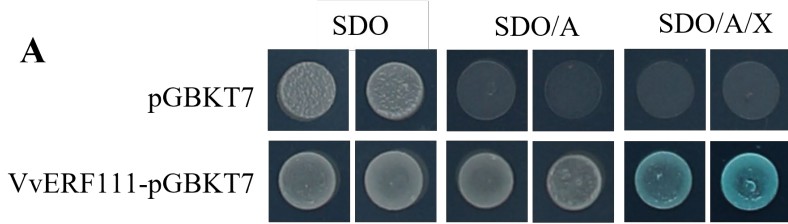

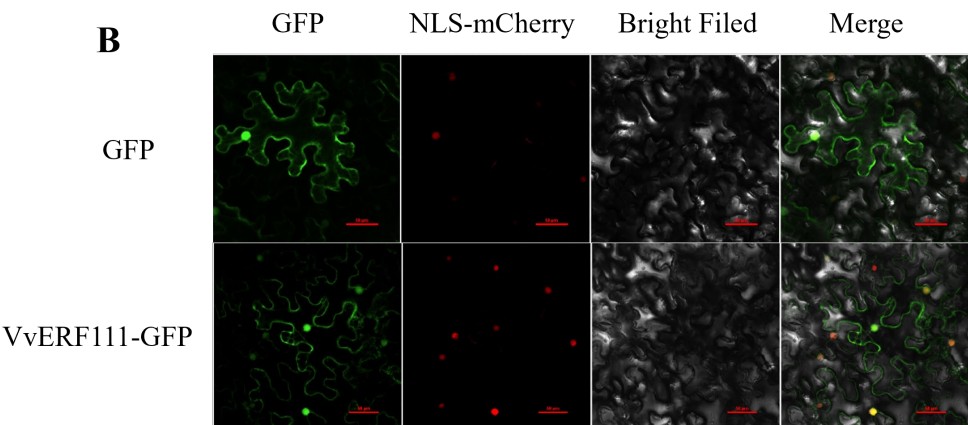

**Figure 3.** Transcriptional activation in yeast and subcellular localization in tobacco of *VvERF111*. (**A**) VvERF111 transactivation activity assay. SDO: synthetic dropout medium; SDO/A: synthetic dropout medium lacking tryptophan and supplemented with Aureobasidin A; and SD/AbA/X: synthetic dropout medium lacking tryptophan and supplemented with Aureobasidin A and X-a-Gal. pGBKT7 vector was used as negative control. (**B**) Subcellular localization of VvERF111 in *Nicotiana benthamiana* leaves, the colored dots represent nucleus. Bright field, white light; merged, combined GFP, and brightfield signals. pBI221-mCherry fusing the NLS (amino acid sequence PKKKRKV) to a red fluorescent protein was used as nuclear-localized marker. Bar = 50 μm.

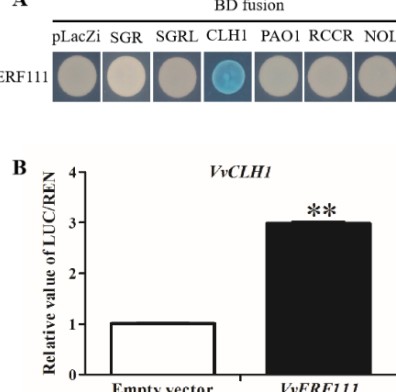

**Figure 4.** Interaction of the VvERF111 protein with the promoter of *VvCLH1*. (**A**) Yeast one hybrid assays to detect the binding of VvERF111 on the promoters of chlorophyll degradation genes. (**B**) Dual-LUC experiment to detect the regulation of the *VvCLH1* promoter by VvERF111, and the pSAK277 empty vector co-transformed with promoter of *VvCLH1* was used as negative control. Data represented mean ± SE from three independent biological replicates; ** represents significance at $p < 0.01$.

To find the potential binding motifs in the promoter of *VvCLH1*, the promoter of *VvCLH1* was truncated into three segments (*p1*, *p2*, and *p3*). Yeast one-hybrid analysis was carried out with *VvERF111*. The farthest promoter fragment, *p1*, could be activated by *VvERF111* (Figure 5A). The *p1* segment was then further divided into six fragments, which were then co-transformed with *VvERF111* into EGY48. The results showed that VvERF111 could bind to fragment *p1–3*, which contained one DRE motif (GGCTG) (Figure 5B). Mutation in the *p1–3* fragment indicated that *VvERF111* binds the DRE *cis*-acting element (GGCTG) of the *VvCLH1* promoter (Figure 5C). We therefore deduced that VvERF111 could bind to the DRE motif in the promoter of *VvCLH1*.

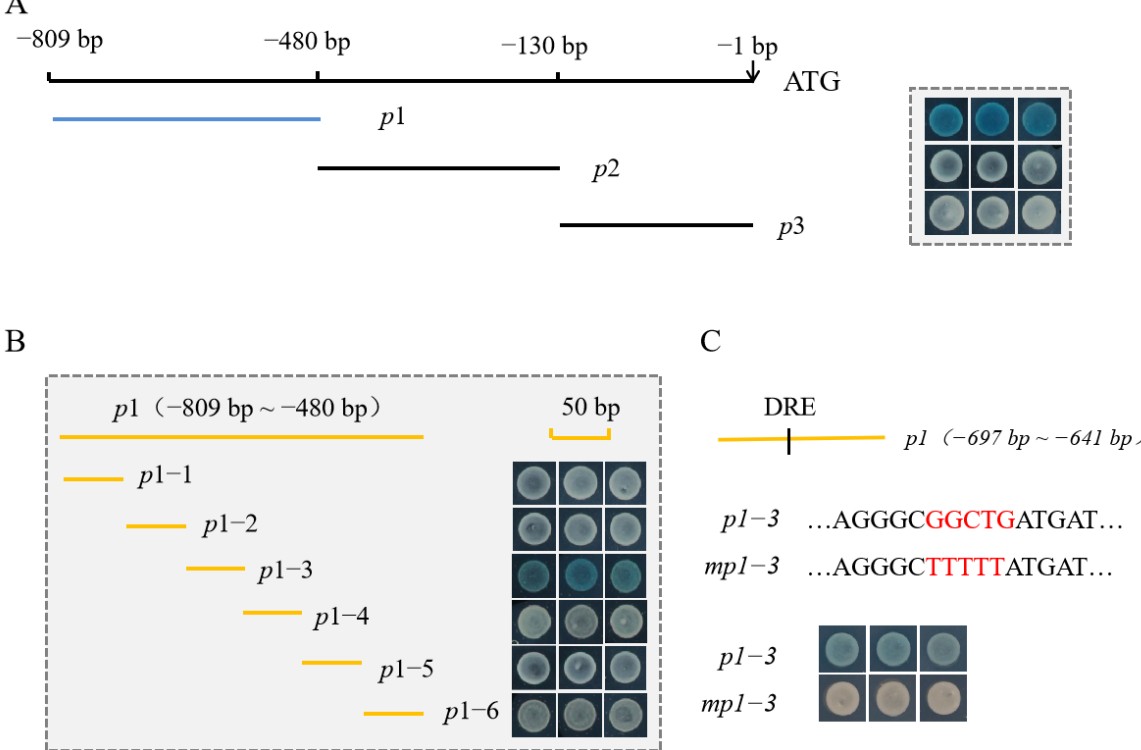

**Figure 5.** Sequential deletion and mutation of the *VvCLH1* promoter to detect the *cis*-element that can interact with VvERF111. (**A**,**B**) A sequential deletion series and (**C**) a mutant of the *VvCLH1* promoter were analyzed in yeast one-hybrid assays expressing *VvERF111.* The blue line represent the segment which VvERF111 binding, and the DRE motif and its mutant indicated in red.

*3.5. Transient Overexpression of VvERF111 and VvCLH1 Promoted Chlorophyll Degradation in Grape*

Since obtaining stable transgenic grape plants would take a long time, the transient overexpression of *VvERF111* and *VvCLH1* in grape leaves was used to determine if *VvERF111* and *VvCLH1* could influence chlorophyll degradation, and *Agrobacterium tumefaciens* cultures containing the *35S-VvERF111-GFP* and *35S-VvCLH1-GFP* expression cassettes were vacuum-infiltrated into grape leaves and observed 3 d later. The expression of *VvERF111* and *VvCLH1* was increased after transient over-expression of the same gene (Figure 6C and left of 6D). The leaves transiently over-expressing *VvERF111* or *VvCLH1* were yellowed compared with the control (Figure 6A), which indicates a lower content of chlorophyll. Transgenic leaves overexpressing *VvERF111* or *VvCLH1* contained significantly decreased chlorophyll content (Figure 6B). In addition, qRT-qPCR analysis revealed that the transcript levels of *VvCLH1* were significantly higher when only *VvERF111* was over-expressed (Figure 6D). This alteration was consistent with our hypothesis that *VvERF111* could regulate the degradation of chlorophyll by regulating the expression of *VvCLH1*.

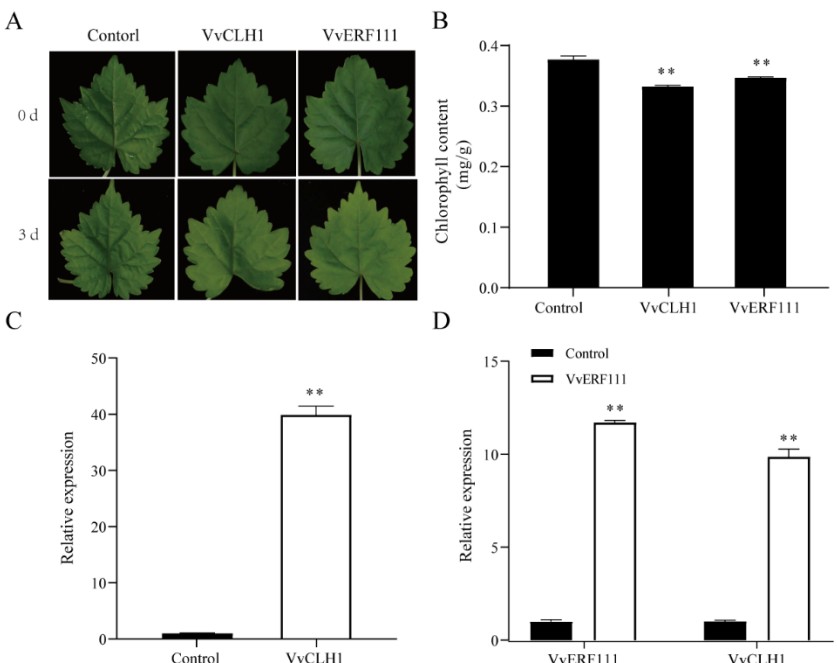

**Figure 6.** Transient overexpression of *VvERF11* and *VvCLH1* in grape leaves. (**A**) Color changes of grape leaves over-expressing *VvERF111* and *VvCLH1*. (**B**) Chlorophyll content in grape leaves after *VvERF111* and *VvCLH1* transient over-expression in grape leaves. (**C**) Expression of *VvCLH1* in grape leaves with transient over-expression of *VvCLH1*. (**D**) Expression of *VvERF111* and *VvCLH1* in grape leaves with transient over-expression of *VvERF111*. Leaves transformed with *35S-GFP* were used as control, data represented mean $\pm$ SE from three independent biological replicates, and ** represents significance at $p < 0.01$.

## 4. Discussion

Color is a critical index for fruit quality, and in grape, the color of both the rachis and the fruit affect consumer choice. Rachis browning in grape affects storage longevity and consumer selection of a grape cluster, decreasing its commercial value. Grape is a non-climacteric fruit; however, ethylene is released from the rachis during post-harvest storage and is an important factor in rachis browning [12,15,22]. Ethylene can induce chlorophyll degradation [12,13,22]; however, the mechanism underlying ethylene-induced chlorophyll degradation remains less studied in grape rachis. The transcriptional regulation of chlorophyll degradation by ERF transcription factors has been widely studied during the degreening of the peel, while the fruit is ripening or being stored after harvest [5,11,30–32]. ERF transcription factors can bind to DRE or GCC-box motifs in promoters of chlorophyll degradation genes, thus regulating their expression [5,12,20]. ERFs are classified into 12 subfamilies [29,33], and VvERF111 belongs to the ERF IX subfamily, which contains 40 members [33]. VvERF111 has the conserved AP2/ERF domain (Figure 2), similar to VvERF95 and VvERF75 in the same subfamily [12]; however, the roles of most genes in this subfamily in grape remain unknown.

Chlorophyll content is decreased during rachis browning and the fruit coloration of grape [12,17,21], while the expression of *VvERF95*, *VvERF75,* and *VvERF17* increases. *VvERF95* and *VvERF75* regulate chlorophyll degradation during rachis browning and fruit ripening, respectively, by binding to the DRE motif in the promoter of *VvPAO1* [12,21], while *VvERF17* regulates chlorophyll content in grape skin by positively regulating the expression of *NOL*, *PPH*, *PAO*, and *RCCR* [17]. During citrus fruit degreening, the expression levels of *CitERF13*, *CitERF6,* and chlorophyll degradation genes increase, while chlorophyll content deceases. *CitERF13* and *CitERF6* trigger chlorophyll degradation by binding the promoter of *CitPPH* and activating its expression [4,20]. During apple peel degreening, *MdERF17* regulates the expression of *NYC* and *PPH*. The binding of ERF17 to chlorophyll degradation-

related genes is affected by the number of Ser repeats [5,34]. The phosphorylation of MdERF17 at residue Thr67 by MdMPK4-14G is necessary for its transcriptional regulatory activity and its regulation of Chl degradation [21]. In banana, the expression of *MaERF012* is closely related to fruit ripening, and MaERF012 activates the promoter of *MaSGR1* during fruit ripening [35]. Together, these studies suggest that ERF transcription factors have conserved functions in controlling chlorophyll degradation in fruit trees.

We previously showed that *VvERF111* showed higher expression levels during rachis browning and that its expression was positively correlated with the expression of chlorophyll degradation genes and negatively correlated with content of chlorophyll [22]. We proposed that *VvERF111* may regulate chlorophyll degradation. The results of yeast one-hybrid and dual-LUC assays suggested that *VvERF111* could positively regulate the expression of *VvCLH1* by binding to the DRE motif in the promoter of *VvCLH1* (Figures 4 and 5), similar to previous reports [12]. *VvERF111* and *VvCLH1* had higher expression during rachis browning [22], and the expression of *VvERF111* could be induced by ethylene and inhibited by 1-MCP (Figure 1). A similar response of *VvCLH1* was detected, which indicated that the expression of these two genes in grape rachis was controlled by ethylene. This is consistent with previous research on the effect of ethylene on chlorophyll degradation [18].

The transient over-expression of *VvERF111* and *VvCLH1* in grape leaves resulted in decreased chlorophyll content compared to the control (Figure 6A,B), which indicated that these two genes had important functions in chlorophyll catabolism. In leaves transiently overexpressing *VvERF111*, the expression of *VvCLH1* was increased, further validating the idea that *VvERF111* activated the expression of *VvCLH1*. In addition to chlorophyll degradation, reduced biosynthesis can also lower the content of chlorophyll. A recent study in tomato showed that *SlERF.J2* regulates chlorophyll accumulation by suppressing the expression of genes related to chlorophyll synthesis [18]. Any potential ERF transcription factor in grape that regulates chlorophyll biosynthesis will be sought in future studies.

Our results revealed that the AP2/ERF transcription factor VvERF111 localized in the nucleus and at the cell membrane, that both *VvERF111* and *VvCLH1* are ethylene-inducible genes, and that *VvERF111* could regulate chlorophyll degradation during rachis browning by binding to the DRE motif in the promoter of *VvCLH1*. This research adds to our understanding of the mechanisms underlying rachis browning in the clusters of table grapes.

**Supplementary Materials:** The following supporting information can be downloaded at https://www.mdpi.com/article/10.3390/horticulturae9040438/s1. Table S1: Primers used in this study. Table S2: Transcriptome analysis of the expression of *VvERF111* and *VvCLH1* during rachis browning.

**Author Contributions:** Conceptualization, D.Z., J.L., Z.L. and J.F.; methodology, D.Z., J.L., J.C. and X.Y.; software, D.Z., J.L., Z.L. and B.T.; validation, D.Z., J.L., X.Z. and W.W.; formal analysis, D.Z., J.L. and Z.L.; investigation, D.Z., J.L. and X.Y.; resources, X.Y., X.Z. and J.F.; data curation, D.Z., J.L. and B.T.; writing—original draft preparation, D.Z. and Z.L.; writing—review and editing, X.Y. and J.F.; supervision, J.C. and W.W.; and funding acquisition, Z.L., X.Y. and J.F. All authors have read and agreed to the published version of the manuscript.

**Funding:** This work was supported by the Key Scientific Research Project of Henan Province Colleges and Universities (Grant No. 21A210021), the National Natural Science Foundation of China (Grant No. 32002017), a Postdoctoral Research Grant in Henan Province (Grant No. 202002054), the Henan Province Outstanding Foreign Scholar Program (Grant No. GZS2020007), and the Natural Science Foundation of Henan Province (Grant No. 222300420457).

**Institutional Review Board Statement:** Not applicable.

**Informed Consent Statement:** Not applicable.

**Data Availability Statement:** All data generated or analyzed during this study are included in this published article and its supplementary information files.

**Conflicts of Interest:** The authors declare that they have no conflict of interest.

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
