# Peer review of "VvERF111 Regulates Chlorophyll Degradation by Activating Expression of VvCLH1, Leading to Rachis Browning in Grape"

_horticulturae, doi:10.3390/horticulturae9040438_

Round 1

Reviewer 1 Report

Please review the comments in the manuscript.

Author Response

General comment: the use of punctuation marks in the manuscript should be improved as there is an excessive use of commas. I only mark a few examples in the text.

Response: Crammer and sentence of this paper has improved by native English speaker.

The discussion should be improved as it was mainly used to recapitulate results.

Response: Discussion was improved.

The conclusion should be presented more clearly.

Response: Additional information was presented in conclusion to make it more clearly.

Reviewer 2 Report

Comments and Suggestions for Authors

Dear Author,

I have an honor to review the manuscript entitled “VvERF111 regulate chlorophyll degradation during rachis browning by activate expression of VvCLH1” a research article submitted to MDPI Journal, Horticulture. Authors of this manuscript identified and characterized plant-specific VvERF111 gene in grape and showed its possible roles in rachis browning. A series of characterization including physical and chemical characteristics, localization, expression analysis and transcriptional regulation have been performed. Overall, the experiments are performed well and the results are convincing. Thus, the presented results takes up an important topic consistent with the profile of the Journal.

-However, even, manuscript is well organized and well described of the conception, I have some suggestions, which might improve the manuscript to make important to the wider audience.

-This article required firm aim of the study that should be underlined precisely and simultaneously and highlight why the rachis brewing analysis and its regulator is important to study in grape.

-Some major and other minor comments are as below

-There are many places where grammar can be improved. I suggest a careful revision by a professional language editing service. Extensive editing of English language and style is required. I've just noted a few here.

-Few suggestions I have mentioned in the main text pdf file. Please check

Title:

According to abstract, rachis browning is result of chlorophyll degradation that regulated by VvERF111, therefore, title may change as

- VvERF111 regulate chlorophyll degradation by activate expression of VvCLH1 leading to rachis browning in grape.

Abstract: -Good organization with results order.

However, need improvement with results representation in full.

-Full form of these genes should be stated in first time (VvERF111, VvCLH1)

-Scientific name of grape should be used

-Keywords:

-Better use alphabetic order

1. Comment in Introduction:

-When using gene name, they should be italic throughout

- At same time, pigments supply nutritions and functional ingredients. Grape is one of the most important fruit trees planted world widely, the goal in table grape storage is to maintain the overall quality of the bunches with the priority being to prevent berry decay.-------------There is something not clear to understand. This type information should be clear with reference for the wider readers.

-Introduction should be more informative and sequential including some more specific findings referencing recent publications.  Rationale to be elucidated for the purpose of the study. Write more information similar to your work.

2. Methods

If available, plant, fruit, age should be used. Need to mention experiment time, season

-It is important to indicate the time of database browsed

Results

-3.2. VvERF111 belongs to ERFIX subfamily and has AP2/ERF conserved domain-----It can be sued first then

---3.1. Expression of VvERF111 and VvCLH1 is affected by ethylene

Not clear message in results. Very less linking between experiments, identified results and morpho-physiological changes of rachis in grape. For each sections of results: Need clear indication why this experiment has been done and in what aspect it justifies the results and prediction.

For example

3.4. VvERF111 binds to the VvCLH1 promoter—What is clue here?? Sub-title should be clearly indicative

Discussion

-Transcriptional regulation of ERF on chlorophyll degradation has been widely studied in plants [5,15,28-30], ERFs can bind-ing to DRE or GCC-box motif in promoters of chlorophyll degradation genes [5,17,31], and ERFs could be classified into 12 subfamilies [26,32], VvERF111 belongs to ERFIX sub- family, and has conserved AP2/ERF domain (Figure 2), similar with VvERF95 in same subfamily [17].-------This sentence should be corrected with informative linking related to identification

More related results should be discussed with recent publications

My Recommendation about this article:

Revision needed. This article may be accepted after major corrections.

Author Response

Dear Author,

I have an honor to review the manuscript entitled “VvERF111 regulate chlorophyll degradation during rachis browning by activate expression of VvCLH1” a research article submitted to MDPI Journal, Horticulture. Authors of this manuscript identified and characterized plant - specific VvERF111 gene in grape and showed its possible roles in rachis browning. A series of characterization including physical and chemical characteristics, localization, expression analysis and transcriptional regulation have been performed. Overall, the experiments are performed well and the results are convincing. Thus, the presented results takes up an important topic consistent with the profile of the Journal.

-However, even, manuscript is well organized and well described of the conception, I have some suggestions, which might improve the manuscript to make important to the wider audience.

-This article required firm aim of the study that should be underlined precisely and simultaneously and highlight why the rachis brewing analysis and its regulator is important to study in grape.

-Some major and other minor comments are as below

-There are many places where grammar can be improved. I suggest a careful revision by a professional language editing service. Extensive editing of English language and style is required. I've just noted a few here.

Response: Thanks very much for your kind suggestion, English language of this manuscript has been improved.

-Few suggestions I have mentioned in the main text pdf file. Please check

Title:

According to abstract, rachis browning is result of chlorophyll degradation that regulated by VvERF111, therefore, title may change as

VvERF111 regulate chlorophyll degradation by activate expression of VvCLH1 leading to rachis browning in grape.

Response: Thanks for your kind suggestion. Title has been changed to ‘VvERF111 regulate chlorophyll degradation by transcriptional activate expression of VvCLH1 leading to rachis browning in grape’.

Abstract: -Good organization with results order.

However, need improvement with results representation in full.

-Full form of these genes should be stated in first time (VvERF111VvCLH1)

-Scientific name of grape should be used

Response: Full form of genes (VvERF111VvCLH1) and scientific name of grape have been added.

-Keywords:

-Better use alphabetic order

Response: Keywords reordered use alphabetic order.

  1. Comment in Introduction:

-When using gene name, they should be italic throughout

- At same time, pigments supply nutritions and functional ingredients. Grape is one of the most important fruit trees planted world widely, the goal in table grape storage is to maintain the overall quality of the bunches with the priority being to prevent berry decay.-------------There is something not clear to understand. This type information should be clear with reference for the wider readers.

Response: “At same time, pigments supply nutritions and functional ingredients.” has been delete from the text, and reference was supplement.

-Introduction should be more informative and sequential including some more specific findings referencing recent publications.  Rationale to be elucidated for the purpose of the study. Write more information similar to your work.

Response: Thanks for your kind suggestion. More information was supplement in Introduction.

  1. Methods

If available, plant, fruit, age should be used. Need to mention experiment time, season

-It is important to indicate the time of database browsed

Response: Thanks for your kind suggestion, experiment time and season as well as age of fruit trees were supplemented.

Results

-3.2. VvERF111 belongs to ERFIX subfamily and has AP2/ERF conserved domain-----It can be sued first then

---3.1. Expression of VvERF111 and VvCLH1 is affected by ethylene

Response: Section of 3.2 and 3.1 has been reordered.

 Not clear message in results. Very less linking between experiments, identified results and morpho-physiological changes of rachis in grape. For each sections of results: Need clear indication why this experiment has been done and in what aspect it justifies the results and prediction.

For example

3.4. VvERF111 binds to the VvCLH1 promoter—What is clue here?? Sub-title should be clearly indicative

Response: Thanks very much for your kind suggestion, addition messages were added in the text.

 Discussion

-Transcriptional regulation of ERF on chlorophyll degradation has been widely studied in plants [5,15,28-30], ERFs can bind-ing to DRE or GCC-box motif in promoters of chlorophyll degradation genes [5,17,31], and ERFs could be classified into 12 subfamilies [26,32], VvERF111 belongs to ERFIX sub- family, and has conserved AP2/ERF domain (Figure 2), similar with VvERF95 in same subfamily [17].-------This sentence should be corrected with informative linking related to identification.

Response: Thanks very much for your kind remind, We have checked the results and correct the reference.

Reviewer 3 Report

This study is about the role of a transcriptional activator namely VvERF111 that can bind to the promoter region of a gene VvCLH1, which encodes for chlorophyllase, an enzyme that degrade chlorophyll and responsible for rachis browning in grapes. The research group also conducted similar studies and found out the relationship of other transcriptional regulators and their respective regulated genes, which also involve in rachis browning. Altogether, this study contributes partly to the discovery and explanation of rachis browning mechanism in grape. And the understanding of this rachis browning mechanism may be helpful in enhancing the grape production and storage.

Specific comments:
1) The manuscript has many minor typo issues. For example, L74,76 “was download”, L83 “were list”, L152 “form”, L208 “Tranaient”, etc.

2) L43-45, the authors should explain why ethylene treatment is carried out in postharvest grape before emphasizing its side effect. Same for 1-MCP and please provide full name for 1-MCP. Readers who is new to the field may not know the function of the treatment.

3) L53-54, “Chlorophyll degradation … browning.” can be deleted or moved to L47. “increased ethylene production” is contradict with the statement at L43 “grape is … has no ethylene release …”

4) L81, “SD-Trp medium” and L82, is SDO same as SD-Trp?. Please use the same naming system for the media as L93 before mentioning their acronyms in bracket.

5) Section 3.1 is confusing. L127-128 and Supplemental Table S2. This finding is from previous transcriptome study? What are the meaning for the setting CK0, CK2, CK4, MCP2, MCP4 in Supplemental Table S2. Please explain clearly. L131-135, this is the current finding for this study using RT-qPCR, right? The author should show the finding in current study first before relating it to the findings from other studies.

6) The resolution for Figure 2B is very poor.

7) L156, strain Y2HGold

8) L174 “The results indicated that”

9) L186 “strain EGY48”

10) L186 and L187 “p3” should be “p1-3”

11) L217-218, change to “the increment of ethylene release during post-harvest storage was an important factor in rachis browning.” Please cross-check the description of this paragraph has the same meaning in introduction (refer to comment #2).

12) L220-225. Split the sentence.

13) L226, please standardize “degreening” to “browning”

14) L248-249, the conclusion should reflect the main finding of the study only, please delete “this suggest that … degradation”.

Author Response

Comments and Suggestions for Authors

This study is about the role of a transcriptional activator namely VvERF111 that can bind to the promoter region of a gene VvCLH1, which encodes for chlorophyllase, an enzyme that degrade chlorophyll and responsible for rachis browning in grapes. The research group also conducted similar studies and found out the relationship of other transcriptional regulators and their respective regulated genes, which also involve in rachis browning. Altogether, this study contributes partly to the discovery and explanation of rachis browning mechanism in grape. And the understanding of this rachis browning mechanism may be helpful in enhancing the grape production and storage.

Specific comments:

  • The manuscript has many minor typo issues. For example, L74,76 “was download”, L83 “were list”, L152 “form”, L208 “Tranaient”, etc.

Response: Thanks for your kind suggestion. Language of this MS has been improved.

2) L43-45, the authors should explain why ethylene treatment is carried out in postharvest grape before emphasizing its side effect. Same for 1-MCP and please provide full name for 1-MCP. Readers who is new to the field may not know the function of the treatment.

Response: Thanks for your kind suggestion. Grape is classified into non-climatic fruit as there has no ethylene release and respiration peak during fruit ripening, however, production of ethylene in the rachis was higher than that in the berry [27], and this information has been added in the text. Full name of 1-MCP was added in the text.

3) L53-54, “Chlorophyll degradation … browning.” can be deleted or moved to L47. “increased ethylene production” is contradict with the statement at L43 “grape is … has no ethylene release …”

Response: Thanks for your kind suggestion. “Chlorophyll degradation … browning.” has been moved to L47. Grape is classified into non-climatic fruit as there has no ethylene release and respiration peak during fruit ripening, however, production of ethylene in the rachis was higher than that in the berry, ethylene treatment enhanced rachis browning while 1-MCP inhibited it.

  • L81, “SD-Trp medium” and L82, is SDO same as SD-Trp?. Please use the same naming system for the media as L93 before mentioning their acronyms in bracket.

Response: SDO is same as SD-Trp, acronyms has been added in the text. Abbreviation was commonly used in yeast two hybrid system, and full name was usually used in yeast on hybrid system, thus, full name was used in L93.

  • Section 3.1 is confusing. L127-128 and Supplemental Table S2. This finding is from previous transcriptome study? What are the meaning for the setting CK0, CK2, CK4, MCP2, MCP4 in Supplemental Table S2. Please explain clearly. L131-135, this is the current finding for this study using RT-qPCR, right? The author should show the finding in current study first before relating it to the findings from other studies.

Response: Yes, this finding is from previous transcriptome study, CK0, CK2, CK4, MCP2, MCP4 means control and 1-MCP treatment at 2 and 4 weeks after harvest during cold storage, CK has changed to control, transcriptome analysis was carried out to identify differential expressed genes during rachis browning. L131-135, is the current finding for this study using RT-qPCR.

  • The resolution for Figure 2B is very poor.

Response: resolution for Figure 2B has been improved.

  • L156, strain Y2HGold

Response: Thanks for your kind suggestion. “Y2H gold” has been changed to “strain Y2HGold”.

  • L174 “The results indicated that”

Response: Line 174 has been changed to “The results indicated that”.

  • L186 “strain EGY48”

Response: L186 has been changed to “strain EGY48”.

10) L186 and L187 “p3” should be “p1-3”

Response: “p3” has been changed to “p1-3” in L186 and L187.

11) L217-218, change to “the increment of ethylene release during post-harvest storage was an important factor in rachis browning.” Please cross-check the description of this paragraph has the same meaning in introduction (refer to comment #2).

Response: Thanks for your kind suggestion, we cross-check the description of this paragraph has the same meaning in introduction and changed to “the increment of ethylene release during post-harvest storage was an important factor in rachis browning.”in L217-218.

  • L220-225. Split the sentence.

Response: Sentence L220-225 has been split.

13) L226, please standardize “degreening” to “browning”

Response: “degreening” has been changed to “browning”.

14) L248-249, the conclusion should reflect the main finding of the study only, please delete “this suggest that … degradation”.

Response: “this suggest that … degradation”has been delete from conclusion.

Round 2

Reviewer 2 Report

Manuscript has been improved. Can be accepted

Author Response

Thanks a lot for you and the Editors’ comments concerning our manuscript entitled “VvERF111 regulates chlorophyll degradation by activating expression of VvCLH1, leading to rachis browning in grape”. 

Reviewer 3 Report

I appreciate that the authors put efforts in enhancing the overall quality and grammar of the manuscript. Unfortunately, the current version of the manuscript appears to have deteriorated in quality and readability, particularly in the introduction section. Besides, there are still many typos present that impacted the overall reading experience. Therefore, I could not accept the manuscript in the present form and I strongly suggest that the authors send the manuscript for professional English editing.

Please refer below a list of typos that I have identified so far, as well as some specific comments for the introduction part. Additionally, it should be noted that the typos mentioned below are not an exhaustive list, and it is recommended that the authors thoroughly review the entire manuscript for errors.

1) L25 "... fruits that is cultivated ...".

2) L26 "... must be carried out ...".

3) L28 "... a green main stem and green individual rachis to each berry are indications of freshness ...".

4) L48 "Grape is a non-climatic fruit as do not have ethylene release and respiration peak during fruit ripening" This sentence is misleading, grape can produce ethylene during fruit ripening, just that it has relatively low or no increase in ethylene production as compared to other climatic fruits, isn't it? Please rewrite the paragraph and you can consider the writing below.

Ethylene is one of the causes of rachis browning. Although grape is considered a non-climatic fruit that does not exhibit a large rise in ethylene production or respiration rate, study showed that the grape rachis can produce more ethylene than grape berries itself during fruit ripening [27], which leading to rachis browning.

5) L54 "Ethylene Response Factor (ERF), is transcription factors within ethylene signaling pathway that associated with chlorophyll levels, either by ... "

6) L59-60
